# Exoskeletons: Contribution to Occupational Health and Safety

**DOI:** 10.3390/bioengineering10091039

**Published:** 2023-09-04

**Authors:** Omar Flor-Unda, Bregith Casa, Mauricio Fuentes, Santiago Solorzano, Fabián Narvaez-Espinoza, Patricia Acosta-Vargas

**Affiliations:** 1Ingeniería Industrial, Facultad de Ingeniería y Ciencias Aplicadas, Universidad de Las Américas, Quito 170125, Ecuador; omar.flor@udla.edu.ec; 2Ingeniería en Diseño Industrial, Facultad de Ingeniería y Ciencias Aplicadas, Universidad Central del Ecuador, Quito 170125, Ecuador; bjcasa@uce.edu.ec (B.C.); jmfuentes@uce.edu.ec (M.F.); 3Unidad de Innovación Tecnológica, Universidad de Las Américas, Quito 170125, Ecuador; santiago.solorzano@udla.edu.ec; 4Carrera de Biomedicina, Universidad Politécnica Salesiana, Quito 170517, Ecuador; fnarvaez@ups.edu.ec; 5Intelligent and Interactive Systems Laboratory, Universidad de Las Américas, Quito 170125, Ecuador

**Keywords:** exoskeleton, occupational health, occupational safety, musculoskeletal disorders

## Abstract

This review aims to characterize the current landscape of exoskeletons designed to promote medical care and occupational safety in industrial settings. Extensive exploration of scientific databases spanning industries, health, and medicine informs the classification of exoskeletons according to their distinctive attributes and specific footholds on the human physique. Within the scope of this review, a comprehensive analysis is presented, contextualizing the integration of exoskeletons based on different work activities. The reviewers extracted the most relevant articles published between 2008 and 2023 from IEEE, Proquest, PubMed, Science Direct, Scopus, Web of Science, and other databases. In this review, the PRISMA-ScR checklist was used, and a Cohen’s kappa coefficient of 0.642 was applied, implying moderate agreement among the reviewers; 75 primary studies were extracted from a total of 344. The future of exoskeletons in contributing to occupational health and safety will depend on continued collaboration between researchers, designers, healthcare professionals, and industries. With the continued development of technologies and an increasing understanding of how these devices interact with the human body, exoskeletons will likely remain valuable for improving working conditions and safety in various work environments.

## 1. Introduction

The work activities of an industry require that, on several occasions, workers perform tasks related to the strain of the body. These activities mainly regard handling loads on the head [1], forced positions requiring squatting positions, and repetitive movements that limit the response and physical resistance depending on the time these tasks are performed. 

Implementing exoskeletons has provided workers with more significant benefits and physical advantages. These benefits allow them to maintain specific positions without requiring great muscular efforts and reduce the risk of injuries or musculoskeletal affectations [2]. Occupational exoskeletons (EXO) have been used as an alternative to reduce fatigue and physical demand in various tasks of workers [3] and are currently employed in manufacturing processes, manufacturing lines [4], and multiple industrial sectors.

The main objective of this research is to provide a comprehensive and up-to-date view of how exoskeletons contribute to occupational health and safety in various industries, examining their current applications, limitations, and the path toward continuous improvement for the benefit of workers. 

This research answers research questions related to the objectives of occupational safety and health (OSH), which involve (1) maintenance of workers’ health; (2) improvement of the working environment and safety; and (3) promotion of a work culture that supports health and safety [5]. Exoskeletons allow the fulfillment of the first two objectives because these devices take care of the physical health of the worker and provide them with greater security in the performance of their tasks. 

Exoskeleton developers have generated devices that have the potential to significantly take care of the health of the worker [6]. In particular, the focus of development remains focused on activities that require high physical demand, with musculoskeletal disorders being one of the most frequent threats [5] due to the uncontrolled handling of loads, poor posture, and repetitive tasks that promote muscle fatigue of the worker [7].

To assist in reducing lower back disorders (LBD), portable back exoskeletons (or “exos”) have been developed to provide ergonomic support, effectively reducing musculoskeletal overexertion [8]. These devices are designed to alleviate strain on the lower back, promoting better working conditions and minimizing the risk of LBD.

Due to the frequency of injuries and risks of musculoskeletal affectations, the construction industry has implemented the use of exoskeletons to mitigate the effects generated by musculoskeletal disorders by reducing physical demand and increasing the capacity of users of these devices [9,10,11]. 

A second primary function of exoskeletons is to provide a means for workers to maintain their non-neutral working postures, limit muscle movement, and reduce human body members’ effort in performing tasks [12,13]. 

There are specific areas of the human body where muscular effort is more remarkable, as in the case of the shoulder, which is subjected to muscle and joint contact forces. Because of this, some developments in passive exoskeletons have focused on assisting while the worker performs quasi-static tasks [14]. 

Another area in which physical effort is concentrated is the lower back. This fact is why sectors of an industry in which load-handling tasks are frequent have implemented the use of exoskeletons to amplify performance and have less impact on the lumbar area of workers [15]. 

The tremendous physical demands due to the use of tools and machine tools by workers is another factor that has driven the development and use of exoskeletons to provide the worker with resistance and physical capacity, reducing the impact of disorders derived from the handling of these tools, including situations of work carried out at height [16,17].

The use and implementation of exoskeletons in workers’ activities should be studied and approached from the point of view of occupational physicians, ergonomists, and occupational health and safety specialists [18]. The experts mentioned above must keep up to date on the developments of exoskeletons [19], their new alternatives, technologies, benefits, and applications for these professionals to make decisions that contribute to the company’s objectives, health, and care of its workers. 

Despite the wide variety of technologies and complexity of developments, the use of single-joint occupational exoskeletons that are portable and that use springs to support the user when handling loads at a considerable height concerning the height of their head is common in several productive sectors [20].

The use of exoskeletons by workers has been well received in activities that correspond to manufacturing environments. According to previous studies, comfort is perceived, and a good adaptation of exoskeletons to the tasks performed provides safety in their use [21].

The structure of the review article is organized as follows: Section 2 addresses musculoskeletal risks in workers. Section 3 focuses on advances in exoskeletons for occupational health and safety. Section 4 explores exoskeletons as support for worker health and safety, addressing assistive exoskeletons, emphasizing those for knee, gait, posture, and low back strain reduction, and arm loading assistance. Section 5 describes the methodology employed in the scoping review, detailing the article selection process using the PRISMA-ScR method. Section 6 presents the discussion, highlighting limitations in using exoskeletons for occupational health and safety and future directions. Finally, Section 7 provides conclusions derived from this review.

## 2. Musculoskeletal Risks in Workers

One factor that influences the generation of musculoskeletal disorders in workers is poor control of activities that require a physical load. The assignment of physical load tasks must consider each job’s morphofunctional level and biomechanical demands [22]. 

Exoskeletons have significantly improved occupational health and safety, improved worker productivity, and assisted aging workers by reducing their risk of musculoskeletal disorders [6]. In addition to this, exoskeletons are also used as haptic devices for training and rehabilitation [23].

The disturbances generated by construction industry workers are the most representative of extraction and mining industries [24]. Studies of multiple musculoskeletal affectations have been carried out in the field of construction and, specifically, for the handling of loads, for which a considerable variety of exoskeletons have been developed [25,26,27,28].

Figure 1 presents a reference to the impact suffered by some parts of the body, such as the back, neck, arms, shoulders, wrists, knees, and ankles, according to the activity carried out according to the activities of the construction industry [29].

According to Figure 1, the back is the most affected in most of the activities carried out by workers in the construction industry, with less impact on the knees and later on the shoulders. Concerning exposure to work-related musculoskeletal disorders (WMSDs), these are more frequent when there are repetitive movements and when the worker performs their activities during kneeling, crouching, stooping, or crawling positions. Concerning the average number of days, we may be unable to work due to injuries, and the shoulder area is the most affected.

It is known that greater force or torque is required to support the movement of some body regions. Electromechanical systems allow driving movements that help mobility in some parts of the body, and some of these systems are used in the development of exoskeletons and prostheses [30].

Figure 2 presents a scheme that identifies the movement of the actuators implemented to support the process of muscle contraction in the most impacted areas, such as the legs, arms, back, and shoulders [31]. Figure 2 presents a scheme of pneumatic or hydraulic actuators that replace the muscles’ activity, benefiting the worker’s activities and providing greater precision, skill, and movement capacity [32].

## 3. Use of Exoskeletons to Improve Occupational Health and Safety

Exoskeletons have been used and implemented in multiple industries as a preventive measure to reduce work-related musculoskeletal stress. The main functional principle of these assistive systems is to transfer mechanical energy to the human body, thereby reducing physical stress on specific body parts. However, there is a controversial debate on whether using exoskeletons prevents work-related musculoskeletal disorders [11]. Implementing exoskeletons in the workplace requires prior actions to determine whether an exoskeleton is strictly necessary and provides a natural solution for the worker and the company. 

Nowadays, companies are implementing exoskeletons as solutions to reduce musculoskeletal disorders in the workplace. According to the European Agency for Safety and Health at Work [33], a percentage of European employees have been identified as working in awkward or strenuous postures. Figure 3 shows the rates according to European countries. The highest values correspond to Spain, France, and Greece. The distribution is not homogeneous in the percentages of the other countries, with significant variability and non-negligible rates. Exoskeletons should be analyzed for cases where the areas allow the incorporation of exoskeletons, contributing to the worker and not hindering their tasks or generating long-term effects due to forced positions in the use of exoskeletons. 

While it is true that exoskeletons can be a tool for improvement, their implementation should not be an isolated action. Still, it should follow the principles and the hierarchy of preventive measures, ensuring that their use firstly helps to eliminate or control the risk factor detected and, secondly, that their introduction in the workplace does not generate new risks to users or third parties, or provoke a response of rejection by those who must use them. In this sense, we recommend reviewing NTP 1163: Exoskeletons II: Criteria for Selection and Integration in the company [34].

As an essential parameter in implementing exoskeletons to support worker safety and health, the time employees spend handling or lifting heavy loads should be considered. Figure 4 shows the percentages of employees (all ages) who spend at least one-fourth of their working time handling heavy loads.

Introducing new technologies in the work environment entails a thorough assessment of Occupational Safety and Health (OSH) for all parties involved. It is essential to have a design that prioritizes human comfort and well-being, as stipulated in the Framework Directive (89/391/EEC) [35]. Due to the variability in the tasks performed and the relationship of new technologies with musculoskeletal diseases, ergonomic conditions are not always evident. It is essential to perform an OSH impact analysis and look for simple solutions to the problem. When all technical measures have been exhausted, such as the use of auxiliary lifting devices or the adaptation of the workspace, it is necessary to address organizational aspects, such as reorganizing work processes. Finally, it is feasible to consider personal measures to safeguard workers.

Exoskeletons can be considered a technical, medical solution and can also be considered as protective equipment. The categorization of exoskeletons depends mainly on their application, design, and purpose of use. Consequently, exoskeletons are only evaluated through a case-by-case analysis. In practice, it is feasible for exoskeletons to be used as technical tools to streamline work processes. However, if their implementation aims to improve the design of a workspace where ergonomic measures are required to prevent injuries due to overloading, they will be considered personal protective equipment. In the future, it will be essential to integrate the evaluation of exoskeletons into the traditional ergonomic approach, focusing on human well-being as they influence work dynamics and organizational aspects.

## 4. Developments in Exoskeletons for Occupational Health and Safety

This section presents multiple advances in exoskeletons designed and built for worker health care and safety. Regarding health, using exoskeletons reduces the overload of effort the muscles are subjected to during the worker’s tasks [36]. In terms of safety, exoskeletons can limit the movements of human body members by avoiding risky positions of multiple body parts [37]. 

The scientific literature reviewed was categorized in Figure 5, types of exoskeletons according to the body part focused on, and kinematic structure categories. 

Figure 5 presents the exoskeletons categorized into two groups; the body-part-focused category presents exoskeletons according to the part of the body for which they have been designed; the developments have focused mainly on the whole body [25,38], upper extremities [39,40], lower extremities, and for specific limbs such as the leg, hand, and arm [41,42]. According to the kinematic structure, the second type has two categories: soft and rigid [20]. Being soft, this exoskeleton type uses flexible elements that adapt to the structure of the worker’s body. The rigid exoskeleton type is a set of elements such as actuators, links, and kinematic joints that allow the user’s mobility.

Figure 6 presents schemes of three categories: according to the action, the type of rigid structure, and the technology used by the actuators that generate movement in the joints of the exoskeletons.

According to the action, the developments have three subcategories: passive [43], semi-active, and active [44]. The category of active exoskeletons refers to systems that employ actuators driven with additional energy, such as pneumatic and hydraulic actuators [45,46]. The semi-active exoskeletons provide energy to support the user’s natural movement, allowing the recovery of the natural positions of the worker’s limbs using mechanisms based on springs and the elastic recovery properties of materials. 

In the case of semi-active and active exoskeletons, the actuators used to drive the joints often require electrical energy. 

Figure 6 presents the non-anthropometric [47] and anthropometric [48] subcategories for the case of the rigid structure type. Non-anthropometric exoskeletons often provide greater load capacities than anthropometric ones since they require rigid actuators that do not adapt quickly to the body shape of the workers. 

The classification-powered technology includes four subcategories: exoskeletons have sets of mechanisms driven by human strength that increase the strength of the arms and legs. A second type is systems that employ hydraulic actuators. A third subcategory employs pneumatic actuators. A fourth category of those that have presented more current developments corresponds to actuators based on muscles formed by multifilaments driven by electrical energy, allowing muscle traction and compression according to the temperature in these polymers. 

## 5. Exoskeletons as Support for the Health and Safety of Workers

Regarding occupational health and safety, developments of exoskeletons have been identified that obey two types of applications: assistance in loading tasks and rehabilitation for the recovery of mobility when workers have suffered disorders and/or injuries. In addition, there are exoskeletons for training that include using haptic devices to provide feedback to the user to correct their postures and train for the tasks to be performed.

### 5.1. Assistance Exoskeletons

According to multiple studies of the most frequent disorders in construction activities, the symptoms and musculoskeletal disorders that present the most recurrence are neck pain; shoulder, upper back, and lower back pain; leg pain; foot pain; general fatigue of the whole body; and hand pain [49,50,51,52]. Below are developments implemented or studied to assist the human body’s members. 

#### 5.1.1. Exoskeletons for Knee Assistance

In order to control and activate the degree of rotational freedom of the knee, developments have been presented that use an elastic actuator operated with a cable. This mechanism allows for the regulation of translation movements with a parallel mechanism of three degrees of freedom, whose movement can be guided by presenting a low inertial resistance for the user [53], guaranteeing safety and robustness in its use. 

Exoskeletons have been designed to provide knee assistance using multimodal actuators. These exoskeletons allow the activation of various modes of operation during movement cycles. In addition, elastic behavior, rigid position, and energy storage and release are controlled [54]. 

Using a mechanical model, an eccentric pulley has been used to achieve multistage nonlinear assistance at different knee flexion angles. The assistance is lower during walking on flat terrain but increases markedly during ascent or climbing by the user [30].

Studies on shape synthesis for exoskeletons that assist the knee have been presented; clinical biomechanical data have been considered to achieve gait cycles, achieving a 45% reduction in maximum actuation force. A four-bar link is used and is intended for use by people with injured knees to support ligaments, tendons, and muscles [55]. 

Knee assistance solutions have been developed that also consider the hip and ankle in multiple gait ranges, and a solution has been proposed for the variable moments that increase with speed. In addition, these developments handle variable torques, which allows for optimizing the assistance benefiting the metabolic energy cost used according to a higher walking speed of up to 1.5 m/s [56].

Subactuated soft exoskeletons have been designed to assist knee extension and plantar ankle flexion for each leg using a single actuator. These developments have made it possible to reduce knee activity by 6.40%. There was evidence of a significant reduction in ankle timing and power [57]. 

To reduce the risk and pain of the knee due to the handling of loads while climbing stairs, an active knee exoskeleton has been proposed that assists in lifting the load by providing an extra pair in the user’s knee joint, which allows for reducing the load of the tendon of the kneecap and the quadriceps in their movement. Using integrated sensors, the force of human–robot interaction is used as a signal to control the device [58].

#### 5.1.2. Exoskeletons for Walking Assistance

The structure and technologies used by exoskeletons for walking assistance have been developed with the support of multiple technologies and types of kinematic joints with good results in the mobile structure [59]. However, developments to achieve a better walking experience improve their performance by using algorithms and control systems that allow the generation of more functional movements and actions for the worker or user of the exoskeleton. Some of these developments are explained below, along with the techniques used.

A technique involving central pattern generators (CPG) has been proposed to enhance the natural movement provided by exoskeletons. This approach enables the application of an adaptable method for modeling the gait trajectory, effectively controlling the movement of lower limb exoskeletons. Using CPG, exoskeletons can provide users with a more natural and intuitive walking experience. These CPGs are synchronized at different joints and updated online in response to the physical behavior of human users to improve their safety and comfort when walking [60]. They also represent an alternative of adaptive central patterns (ACPG) that lack safe interaction and compatibility with human users, for which they consider the adaptive disturbance of limited gain in time. This control technique allows us to better estimate human–robot interaction (HRI) with CPG [61]. Using these systems allows for extending the walking time considerably, reducing fatigue in the user.

Exoskeletons for lower extremities have been developed and evaluated to decrease metabolic cost, movement strategy, and muscle activation under repetitive load lifting and gait conditions. From measurements of energy expenditure, kinematic factors, and muscle activity in 11 healthy men in periods of 5 min for repetitive loading and walking, it was concluded that metabolic costs were reduced by 17%, which required less generation of physical work by the worker [37].

Military workers who frequently walk and handle loads are prone to musculoskeletal injuries and disorders [62]. These activities have benefited from the use of exoskeletons. It was evidenced that in the first 8 days of use of walking assistance exoskeletons, users increased their metabolic cost when walking; however, after this time, when they became accustomed to using the device, this cost was reduced [63]. 

Two types of exoskeletons differ in their application field: assistance and augmentation exoskeletons. Assistive exoskeletons are commonly used to assist or replace affected limbs or joints of users in order for them to restore their movements, increasing their independence and improving their quality of life. Augmentation exoskeletons are used in industries to improve ergonomics, reduce risks of exposure to demanding working conditions, prevent work accidents, and reduce workers’ acute physical stress and stress by increasing their physical efficiency [64].

Exoskeleton walking assistance also incorporates chair support to avoid risks of muscle fatigue in the lower extremities caused by standing for prolonged periods. According to studies, prolonged standing has been associated with an increased likelihood of heart disease [65]. An alternative called ChairX has been proposed, which assists the worker in sitting with the possibility that they can also walk between workstations; its flexibility allows adjustment at various heights to support ergonomics at work. This alternative provides stability, reduces risks of musculoskeletal injuries, and improves the quality of work and safety of the worker in their work [66].

For assistance during level walking, climbing, descending stairs, and up and down ramps, portable exoskeletons have been developed that provide adequate electrical assistance [67]. A rapid method of gait detection based on a set of body sensors has been proposed; the control system of this alternative allows for the provision of torques of assistance necessary to boost human movements more naturally and immediately [68].

#### 5.1.3. Exoskeletons for Posture Assistance

In construction, musculoskeletal disorders represent one of the biggest problems for workers. The way and frequency with which loads are handled without adequate control often cause affectations in the lower back. It has been shown that using exoskeletons reduces flexion in the back, increasing flexion in the hip.

Although there are developments implemented in industries to assist the posture of workers, it is considered that the deployment, widespread adoption, and potential of these alternatives are still limited and depend on multiple factors such as effectiveness, the type of task to be performed, and the body postures that require support [69].

Another aspect to be solved with the use of exoskeletons is the control of the balance of workers while they handle loads. Solutions have been proposed to achieve equilibrium according to the exoskeleton type, and loads handled [70]. 

Exoskeletons have been studied that assist workers whose task is to decontaminate nuclear power plants after disasters, for which force sensors have been implemented in the foot area so that the control system can determine the angles with which they move the user’s limbs and activate the motorized exoskeleton to assist the movement of the joints in a smoother and controlled way [71,72]. For the control of the posture of the lower extremities, alternatives have been proposed that allow for estimating, with 72.5% compliance, the deviation of the posture between the exoskeleton and the position of the knee through the use of the PO-MOESP algorithm [73].

More effective control has been achieved to provide electrical assistance to the exoskeleton actuators depending on the intention of movement of the users; this control method is based on the use of electromyography (EMG), with which the performance of the exoskeleton used for upper extremities has been evaluated, whose data for control have been obtained from the position of the user’s wrist and the extreme tip of the exoskeleton [74]. Some developments employ adaptive algorithms based on passivity for the upper extremity assistance exoskeleton that allows for the use of different user parameters and an adjustment to the biomechanical conditions of each person [75].

As a contribution to preventive measures in tasks that require body movement with an inclination of the trunk, hip exoskeletons have been designed to allow for stabilizing the position under conditions of external disturbances, which has corroborated that the use of the exoskeleton reduces, by 40%, the forward inclination of the trunk, reducing the risk of muscle injuries [76].

For people with disabilities in their lower extremities, exoskeletons on wheels have been developed whose novel design supports the postural transition in the change of movement from sitting to standing and vice versa, providing a greater degree of user independence [77].

In order to reduce fatigue and muscle pain in the legs due to prolonged work in standing and squatting positions, an exoskeleton with a multilink mechanism based on redundant rods has been designed that avoids interference between the exoskeleton and the leg when walking. Based on the behavior of the crank–crank mechanism, the exoskeleton design has been given viability, managing to transmit body weight to the ground when the user feels supported by the exoskeleton [62]. 

#### 5.1.4. Exoskeletons for Less Effort in the Lumbar Area

The injuries that influence the most in quality of life and reduce life expectancy in workers correspond to spinal cord injuries and disorders [63]. 

Developments in the field of lumbar activity assistance exoskeletons include the participation of multiple disciplines such as robotics, mechanisms, bionics, control theory, communication technology, and information processing technology, which allows for integrating solutions to improve the control of the exoskeleton response to facilitate user activities [64].

Assisting augmentation exoskeletons for people with spinal cord disorders or sequelae can get the worker to handle greater loads and achieve the activities that a person would normally do without any considerable affectation; however, the continued use of an exoskeleton for this purpose can promote bone loss and risk of fragility fracture, as is the case with people who constantly use wheelchairs. In order to provide a complete solution, an exoskeleton must allow adequate and necessary mobility so that the person is not affected by downstream effects arising from the use of exoskeletons [65]. 

For adequate handling of loads at considerable heights, the Aerial Porter (APEx) exoskeleton has been created, which consists of a motorized system mounted on the hip and attached to an adjustable vest, according to the behavior of the user; this system activates the motors, which provide a torque of 30 Nm [66]. With the help of electronic control strategies, solutions that facilitate pushing, pulling, and transporting loads and materials with the help of active and partially passive exoskeletons [67] have been designed.

With the use of whole-body powered exoskeletons (WB-PEXO) and during the performance of static load loading and transfer tasks with and without an exoskeleton, a reduction in reduced average levels of muscle activity in the back (42–53% in thoracic and 24–43% in lumbar regions) and legs (41–63% in knee flexors and extensors) has been verified when handling loads of 10 to 15 kg [68]. 

The evaluation of passive exoskeletons during the repetitive lifting of loads in squat positions, crouching, and in orientations such as frontal/symmetrical and lateral/asymmetric orientations has allowed us to measure the extensor muscle activity of the trunk and hip. Exoskeletons were determined to have limited influence on muscle responses; the device supports the hip extension and requires tight body posture during lifting [69]. In addition to this, exoskeletons (SPEXOR) designed for the prevention of low back pain and vocational reintegration have been developed that focus on improving the transfer of load from the extremities to the arms and legs, avoiding low back pain in the user and providing vocational rehabilitation according to their daily tasks [70]. 

#### 5.1.5. Exoskeletons for Arm Loading Assistance

Work performed on the worker’s head (OHW) poses significant risks that can lead to musculoskeletal disorders in the shoulder area. Upper extremity exoskeletons (EXOULs) have been developed to reduce these risks. This type of development seeks to control muscle activity in the postural muscles and upper extremities and balance control during the performance of OHW. Exoskeleton designs must adapt their mass in implementation, balance, and support torque to optimize the use of these devices [70].

With the passive shoulder exoskeleton called Exo4Work, activity, muscle fatigue, and subjective experience during simulated work on the head without overload have been evaluated. Reduced muscle activity and delayed development of muscle fatigue have been shown during OHW [71].

The excessive muscle and joint contact forces generated in the shoulder area that can cause musculoskeletal disorders have been effectively reduced using passive shoulder exoskeletons when performing quasi-static tasks. The evaluation of these advantages has not contemplated sequelae in joints and neighboring areas such as the elbow, lower back, hip, and knee. Further study is required around the assistance versus adaptation of the exoskeleton to the user and its effects on musculoskeletal loads [14]. Insufficient information has been presented in studies on when, what type, and how long users can use robotic exoskeletons without harming a worker [72].

With torsion springs, passive arm exoskeletons have been developed to help workers lift loads or transport at height. The assistance generated with the springs provides a pair of assistance to the elbow and wrist joints that allow us to work with loads of up to 8 kg effectively. An advantage of this exoskeleton design is its low weight, low cost, and ease of use, as it is easy for workers to activate and deactivate the action of the springs at any time [73].

A two-degree-of-freedom exoskeleton incorporating a dual motor–tendon actuator has been designed and tested to assist elbow and wrist movement. A rack gear joint is used to convert the rotational motion of the motor into linear motion, which allows for two types of flexion/extension and pronation/supination movement [72]. 

Figure 7 shows, in a circular dendrogram, the references of the revised documents that correspond to evaluations of the results of active and passive exoskeletons according to the body part for which they assist and highlights the advantages for occupational health and safety. The categories are observed according to the previous sections, and five categories are highlighted for the assistance exoskeletons: arm, posture, knee, lumbar area, and walking. 

Exoskeletons are often studied based on their collective impact rather than individual components in the context of their contributions to occupational health and safety. Therefore, various commercial, experimental, and prototype developments are outlined in Table 1. This table presents an overview of commercial exoskeletons along with their specific functionalities and intended objectives for which they were designed. By categorizing exoskeletons, researchers can comprehensively understand their diverse applications and potential benefits in occupational health and safety. The exoskeletons in Figure 1 are shoulder–arm, full-body, leg, and back assists. 

## 6. Methodology

A systematic review was conducted to compile this document by thoroughly examining the scholarly literature, including journal articles and conference papers from reputable scientific databases and repositories [78]. The most relevant articles published between 2008 and 2023 were extracted from IEEE, Proquest, PubMed, Science Direct, Scopus, Web of Science, and other databases. The PRISMA-ScR checklist (Table A1 in Appendix A) was used in this review, and a Cohen’s kappa coefficient of 0.529 was applied, implying moderate agreement among reviewers; 91 primary studies were extracted from a total of 344. The search strings used for information retrieval are presented in Table 2, which illustrates the specific search terms employed during the research process. This approach ensured a comprehensive and rigorous analysis of the available literature. 

Five research questions were posed for extracting information in the selected documents and from which reference information could be extracted. The research questions posed were RQ1. What types of exoskeletons have been addressed in the studies?; RQ2. What are the applications of exoskeletons for occupational health and safety?; RQ3. How do occupational tasks influence the musculoskeletal affections of the worker?; RQ4. What are the contributions of exoskeletons to occupational health and safety?; and RQ5. What are the limitations that arise in the use of exoskeletons for occupational health and safety?

The quality of the articles considered was evaluated considering the criteria shown in Table 3. In addition, publications corresponding to the last 10 years were considered.

For the search carried out, the following keywords were used: exoskeletal, safety, health, and occupational workers. A related scientific literature search was conducted using Web Of Sciences, ProQuest, ScienceDirect, SCOPUS, PubMed, and IEEE Xplore databases. From the identification of 344 articles, 75 works were obtained that provided relevant information to carry out this review article; the workflow carried out using the PRISMA^®^ methodology is shown in Figure 8.

## 7. Discussion

In the last 10 years, there have been more frequent developments in the field of exoskeletons for use in industries. Robotic exoskeletons have become more available and of interest for implementation in industrial processes, especially construction. The variety of exoskeletons also now has the potential to be successfully used for personal use at home and in the community [75].

Concerning RQ1, various models, technologies, and types of exoskeletons have been identified that have been categorized by their type of assistance in exoskeletons for the knee, walking, posture, the lumbar area, and the arms [20,25,34,35,36]. Classifications have been proposed according to the action performed by exoskeletons [39,40,41,42], structure types [43,44], and actuator technologies used to provide assistance and support to the user [45]. 

Articles have been found with information responding to RQ2 that describes applications of exoskeletons in the field of occupational safety in the performance of tasks related to the handling of moderate loads, which allows for maintaining posture in uncomfortable positions and mutual assistance devices that reduce the force they perform during tasks [1,5,15,20]. The devices used in most industrial applications are passive such as the examples presented in Table 1. 

The tasks that generate the most impact on the development of musculoskeletal affectations correspond to the performance of construction tasks responding to RQ3. The construction field [19,20,31,32,33] and specifically high-load tasks are the situations that cause the most significant interest of exoskeleton developers and researchers since tasks in these conditions provide more significant risk factors in workers [9]. Within the various tasks a construction worker performs, some tasks require greater demand, including painters, carpenters, and plumbers, who are exposed to more significant risks of musculoskeletal disorders, WMSDs (illustrated in Figure 1). The influences of exoskeletons on metabolic muscle activity have been described, and the parts of the body that have benefited or been alleviated with these devices during work activities have been identified.

About RQ4, knowledge gaps are glimpsed in studies carried out with sufficient time for the use of exoskeletons by workers. Although significant advantages have been identified in repetitive and loading tasks, no literature analyzes the muscle areas close to those that interact with the fastening elements of the devices. Exoskeletons and exosuits (EXOs) are designed to protect workers by reducing the risk of exertion injuries and muscle fatigue during work. Little of the scientific literature studies the effects in neighboring areas of the human body that exoskeletons have served; by modifying their activity and energy consumption, some muscles will have a greater or lesser effect on the other muscles or structures with which they interact [14,16]. The usability of EXOs in construction has not yet been sufficiently studied scientifically [76]. 

The limitations to which RQ5 refers have been identified in evaluations of the implementation of exoskeletons for workers of industries manifested with discomfort and movement restrictions, which have been the reason for new studies that propose alternatives to provide improvements in assistance, a torque increase automatically according to the intention of movement, user behaviors, or movement patterns. No studies address, in sufficient depth, aspects such as the optimal use of exoskeletons to perform work tasks [3].

With the implementation of exoskeletons in fundamental tasks within an industry, user perception and usability have been evaluated several times [21]. Although there is still much to study in this aspect of usability and ergonomics in the use of exoskeletons, it has been shown that in the first days of the use of these devices, users present discomfort that must be overcome in an adaptation stage, after which, the worker will use the exoskeletons more comfortably. A comprehensive regulation, technical note, or reference guide addressing the ergonomic aspect of user–exoskeleton interaction is necessary [1]. In the scientific literature, recommendations and consensus statements have been identified on applying exoskeletons in the workplace to reduce and prevent musculoskeletal diseases [18]. 

Continued use of exoskeletons can result in bone fragility due to a lack of limb activity [16]. The associated risks must be thoroughly investigated before further developing these devices. A lack of attention has been observed in the psychological aspect of users [12]. This aspect is crucial, considering the time they spend with exoskeletons in work activities and the movement restrictions they experience. Future research should focus on better understanding users’ adaptation and interaction with exoskeletons and identifying short- and medium-term detrimental factors in their physiology [79].

A word cloud has been made with the keywords from the reference documentation. The image in Figure 9 considers the words exoskeleton, occupational, and musculoskeletal with more remarkable recurrence, followed by human, health wearable, and device. This chart highlights developments and documentation on using exoskeletons in an industry if strongly considering user health over efficiency and economic benefits for the industry. The technological aspect in terms of the participation of robotics to assist is also highlighted.

### 7.1. Limitations on the Use of Exoskeletons for Occupational Health and Safety

The limited sample size and the selection of participants from a specific geographic and work environment may affect the generalizability of the results to different contexts. In addition, socioeconomic heterogeneity may influence the acceptance of new technologies, such as exoskeletons [2].

A lack of practical experience with exoskeletons may make assessing their impact on workers difficult. Participants may struggle to address the expected impact on surgical workers without directly experiencing exoskeletons [10].

Many investigations show clear limitations in methodological quality, as they selectively select certain work activities for an analysis and do not comprehensively address work demands in various fields [11].

Exoskeletons may limit functional performance in tasks that do not involve load manipulation. The exoskeleton may hinder tasks such as walking that require hip flexion, which may also increase metabolic costs [47].

The effects of exoskeletons may depend on the specific task demands and working conditions. The effectiveness of an exoskeleton may vary depending on the task being performed [19].

Most evaluations of exoskeletons focus on short-term effects, especially in laboratory settings. This case may be due to resource and feasibility constraints, which may not fully capture long-term impacts [7].

Results obtained in studies limited to certain types of agricultural workers may not be generalizable to other forms of agriculture or agricultural workers with different working conditions and demands [5].

Measurements based on recognized standards may have limitations, such as assessor-dependent results and poor correlations with direct physiological measurements [8].

Discomfort and a lack of standards and novelty are challenges in exoskeleton design. Discomfort may hinder their acceptance and use in industrial settings, limiting their applicability [47]. The lack of standards and the novelty of exoskeleton technologies make quantitative biomechanical risk assessment and the application of traditional methods difficult.

Exoskeletons can be expensive, which may restrict their adoption in industries with limited budgets [62]. In addition, some studies mention that exoskeletons are customized and expensive [61]. To be effective, exoskeletons must fit each user correctly, which can be challenging due to differences in body morphology [78]. In some cases, exoskeletons may require complex anthropometric adjustments [66].

Prolonged use of exoskeletons can be uncomfortable for workers, especially if they are ill-fitting or interfere with natural movements [65]. The interaction between the exoskeleton and the human body can cause discomfort where they interact [72]. Some exoskeletons may restrict worker mobility, which can be problematic in industries where rapid or complex movements are essential [78].

Workers must be adequately trained to use exoskeletons effectively and safely. This training can take time and resources. Exoskeletons require regular maintenance to ensure that they function correctly [78]. This situation can increase costs and operational complexity.

User acceptance of the device is critical to its success [71]. The interaction between the user and the exoskeleton, as well as the subjective perception of the device, are essential aspects that need to be addressed.

Exoskeletons may be more effective in specific tasks and less effective in others. For example, it has been observed that exoskeletons may limit functional performance in tasks that require hip flexion, such as walking [64]. Exoskeletons may have technical limitations, such as being heavy and bulky, making them difficult to use and transport [66]. 

There is a risk that exoskeletons may cause discomfort, pressure on unwanted parts of the body, changes in the center of gravity, and other health problems, such as skin irritation and allergic reactions [76].

Although exoskeletons show advantages in improving occupational health and safety, the limitations mentioned above should be carefully evaluated by employers before implementing these devices in the workplace. Each exoskeleton has unique characteristics and considerations, so a comprehensive evaluation of its fit, usability, and effectiveness in specific work contexts is needed.

### 7.2. Future Directions

The future of exoskeletons holds promising prospects, marked by ongoing research and development endeavors aimed at enhancing their functionality, comfort, and efficacy across diverse occupational domains [61]. Expert opinions and studies present various avenues for the forthcoming evolution of exoskeletons.

Efforts are being directed toward implementing exoskeletons in surgical settings to mitigate musculoskeletal discomfort among surgical teams [63]. Comparative evidence could bolster the feasibility and effectiveness of exoskeletons in this context.

A prudent approach involves delving deeper into using exoskeletons to prevent musculoskeletal injuries through randomized controlled trials. This approach would furnish robust data regarding exoskeletons’ efficacy in mitigating occupational hazards [60].

To comprehensively understand exoskeleton applicability, we advocate for research encompassing diverse samples based on age, gender, health status, and occupational tasks [11]. This approach would furnish a holistic view of exoskeleton effectiveness in varying work environments.

Anticipated advances in lightweight materials, sensors, and actuators are poised to facilitate the creation of more sophisticated and comfortable exoskeletons. This progress may usher in the next generation of devices featuring enhanced natural movements and superior adaptability to the human body [8].

Incorporating artificial intelligence and augmented reality technologies can potentially elevate exoskeleton functionality and efficacy. This integration allows for improved interaction and intuitive interfaces between exoskeletons and users.

The future envisions exoskeletons incorporating predictive algorithms for optimizing assistance control [46]. Such innovation would curtail energy consumption and safeguard users’ musculoskeletal well-being.

Advancements in technology are expected to render exoskeletons lighter, more user-friendly, and affordable. Robust, ergonomic design and heightened usability are pivotal for seamless integration into work environments.

Seamless integration into work processes is on the horizon, aligning exoskeletons more closely with workers’ needs. This alignment promises enhanced synergy between exoskeletons and occupational tasks.

A future characterized by continual enhancements in design, functionality, comfort, and efficiency lies ahead for exoskeletons. As technology, materials, and scientific knowledge advance, these devices are poised to remain indispensable tools for curbing musculoskeletal risks and elevating occupational health and safety standards across diverse workplaces.

### 7.3. Environmental Impact in the Manufacturing of Exoskeletons

The manufacturing and development of robotic exoskeletons have generated a paradigm shift in various sectors, encompassing healthcare, military, and industrial applications. As these technological advances continue to proliferate, it becomes crucial to consider their impact on the environment to know whether the production of these solutions will contribute to current environmental challenges [80].

In the design process of exoskeletons, a fundamental part is the selection of the energy source, an aspect of notable influence on the environmental footprint of these devices. Electric exoskeletons exhibit a relatively low environmental impact, primarily when powered by renewable energy sources. They do not emit directly and are highly energy efficient. However, the production and disposal of their batteries can have environmental implications. Battery manufacturing often involves the extraction of rare earth metals, which can lead to habitat damage and water pollution. In addition, improper disposal of used batteries can release hazardous materials into the environment.

Hydraulic and pneumatic exoskeletons could have a higher environmental impact. These systems often require fossil fuels to generate the necessary pressure, leading to carbon emissions. In addition, the production of hydraulic fluids and compressed gases can contribute to environmental pollution.

For example, the use of exoskeletons could reduce the need for heavy machinery in the industrial field, resulting in lower energy consumption and emissions. They could also improve occupational safety and productivity, reducing injuries and their associated environmental impact. In health, exoskeletons could provide mobility for people with physical disabilities, reducing dependence on motorized wheelchairs and other devices. This situation could result in significant energy savings and reduced carbon emissions. In addition, promoting independence could reduce the demand for resource-intensive care facilities.

Several strategies are feasible to maximize the environmental benefits of robotic exoskeletons. First, using renewable energy sources to power these devices should be prioritized. This instance could include integrating solar panels into the design or developing more efficient batteries. Second, the lifespan of exoskeleton components should be extended through more efficient design and maintenance. Finally, end-of-life management strategies should ensure environmentally friendly recycling or proper disposal of exoskeleton components [80].

The potential of exoskeletons to promote sustainability is significant. Through the careful choice of energy sources and implementation of strategies to reduce waste and emissions, it is feasible to harness the benefits of this technology in a manner aligned with our environmental goals. As we continue to innovate and refine these devices, the potential for a more sustainable future is increasingly within our reach.

## 8. Conclusions

Multiple designs and developments of exoskeletons have been implemented for commercial and academic purposes worldwide. The morphology of these devices and their functions provide real advantages. They contribute to fulfilling industry workers’ occupational health and safety objectives, emphasizing the construction and manufacturing area where heavier loads and conditions of high physical demand are handled. However, these conditions have not been thoroughly studied.

Most studies and advancements in exoskeletons for occupational tasks have focused on providing shoulder and arms, full-body, leg, and back assistance. These systems are implemented in industrial sectors where some studies have gathered information on usability and user perception. On the other hand, academic studies and developments utilize test benches that enable the independent evaluation of muscular activity and interaction forces. Therefore, to more accurately evaluate the interaction between the worker and the exoskeleton, it is necessary to consider natural and task-specific situations in the workers’ daily routines. Additionally, it is crucial to consider the implications of prolonged usage of these devices to generate more comprehensive and significant studies.

Uncertainties exist regarding the consequences of prolonged exoskeleton use in areas near the joints they support, including the effects of interaction, adaptability, optimization, and impact. It is essential to thoroughly study these factors to enhance the utilization of exoskeletons in work environments, mitigating potential health consequences for workers that may not be readily apparent.

The design and evaluation of exoskeletons require the multidisciplinary participation of doctors, biomedical experts, physiotherapists, mechanics, electronics, and control specialists. The physical aspect of these exoskeletons, in terms of their movement, has undergone significant development and applied technologies. Optimization and improvements have been achieved by implementing automatic control strategies determining the user’s movement intention, behavior, and repetitive movement patterns. This analysis allows for activating and deactivating the exoskeleton’s actuators, initiating movement at the appropriate time, and achieving a more natural motion for the device user.

## Figures and Tables

**Figure 1 bioengineering-10-01039-f001:**
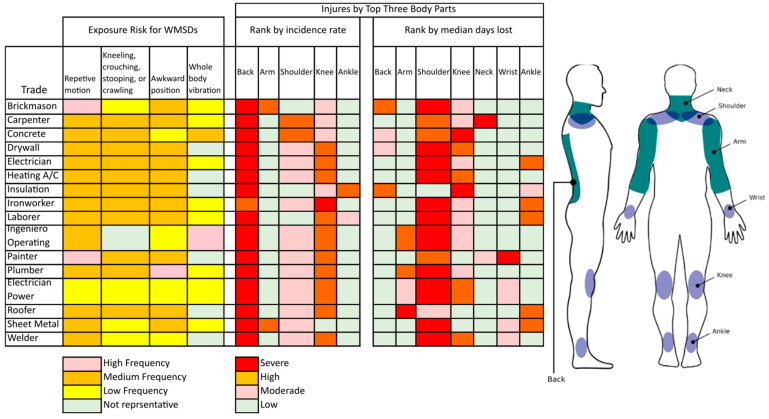
Risks of exposure to disorders and injuries to body parts in construction trades.

**Figure 2 bioengineering-10-01039-f002:**
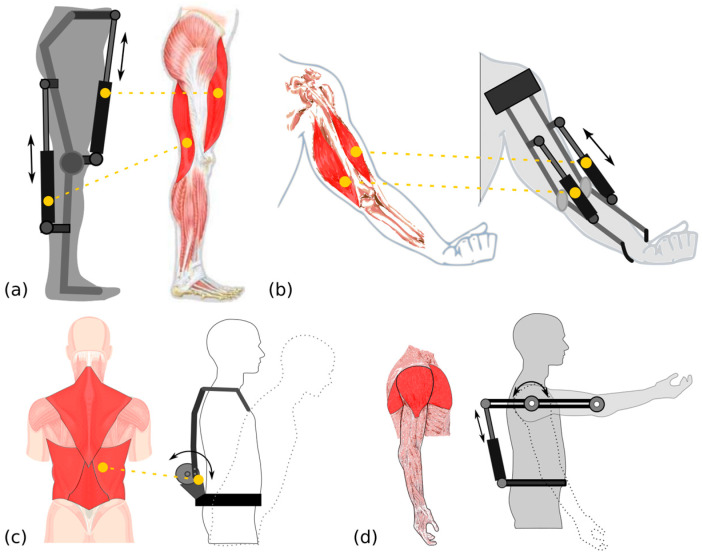
Scheme of assistance of exoskeleton actuators in muscle movement for the (**a**) leg, (**b**) arm, (**c**) back, and (**d**) shoulder.

**Figure 3 bioengineering-10-01039-f003:**
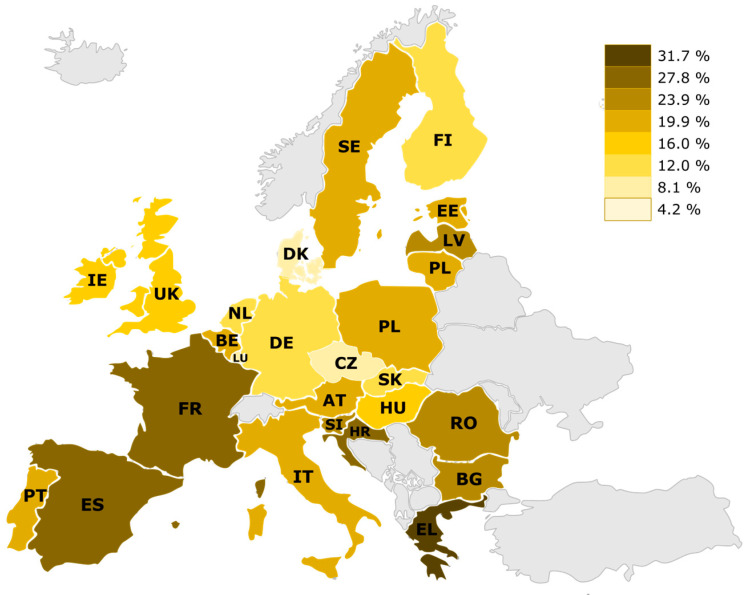
Percentage of workers performing activities with strenuous or forced postures in European countries.

**Figure 4 bioengineering-10-01039-f004:**
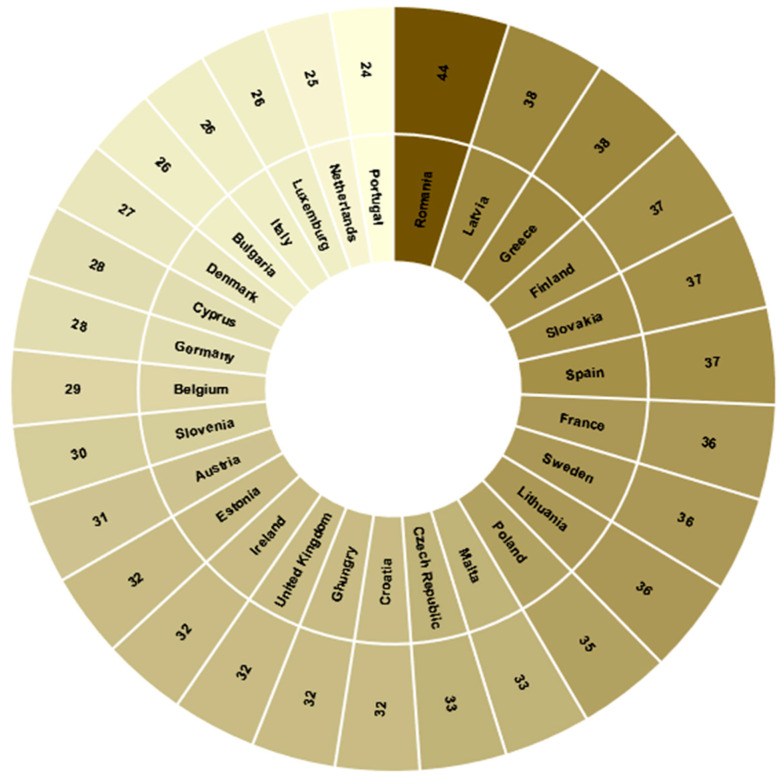
Percentage distribution of employees from European countries that manipulate heavy loads during a quarter of their working day.

**Figure 5 bioengineering-10-01039-f005:**
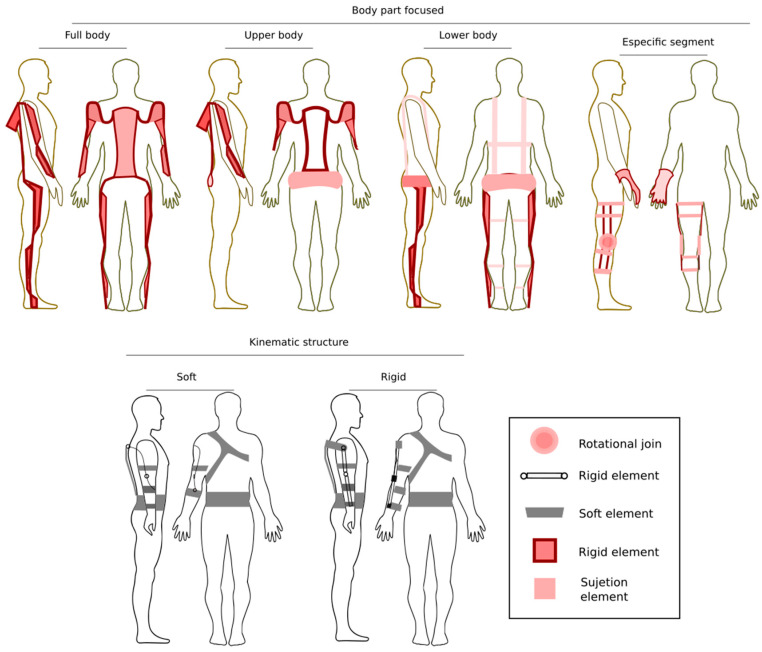
Types of exoskeletons according to body part focused on and kinematic structure.

**Figure 6 bioengineering-10-01039-f006:**
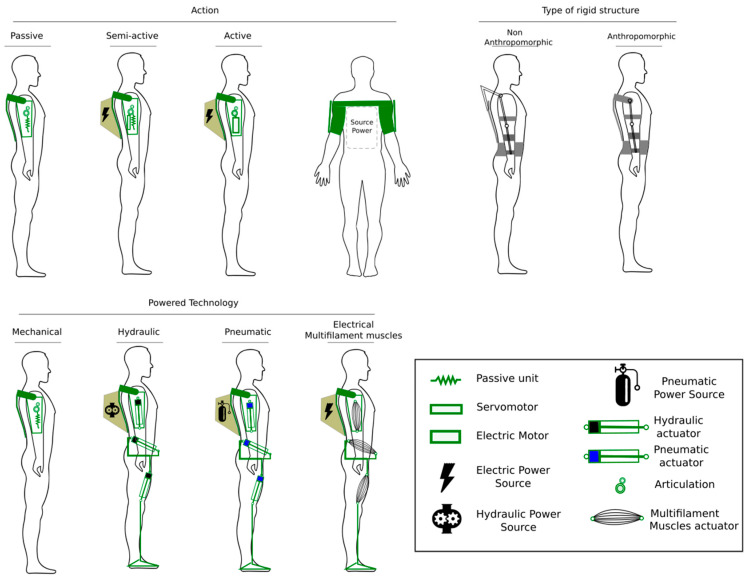
Types of exoskeletons according to their action, type of rigid structure, and technology according to the actuators used.

**Figure 7 bioengineering-10-01039-f007:**
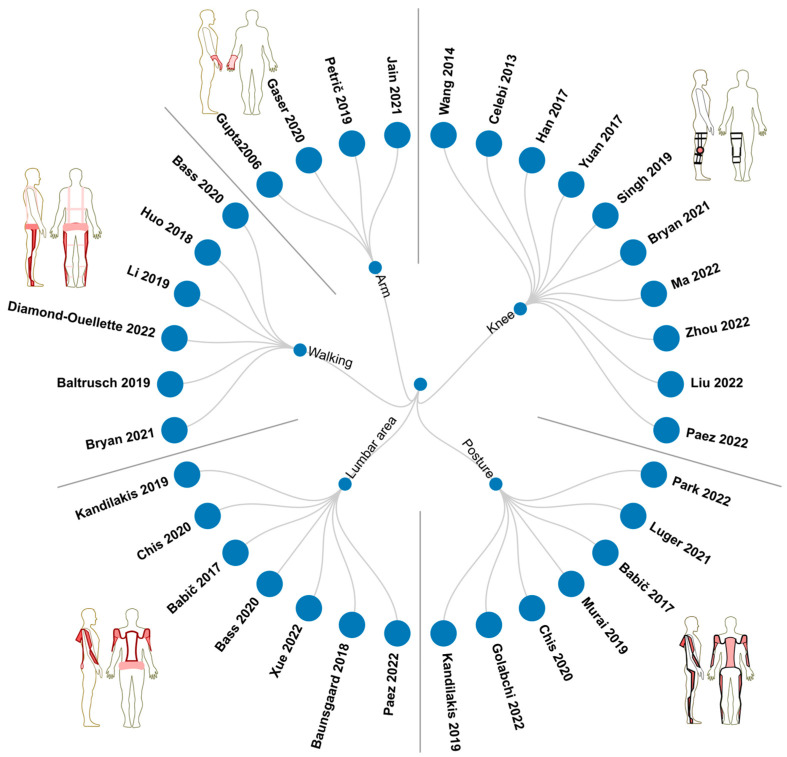
Circular dendrogram of related studies on exoskeleton developments and their contributions to occupational health and safety of workers.

**Figure 8 bioengineering-10-01039-f008:**
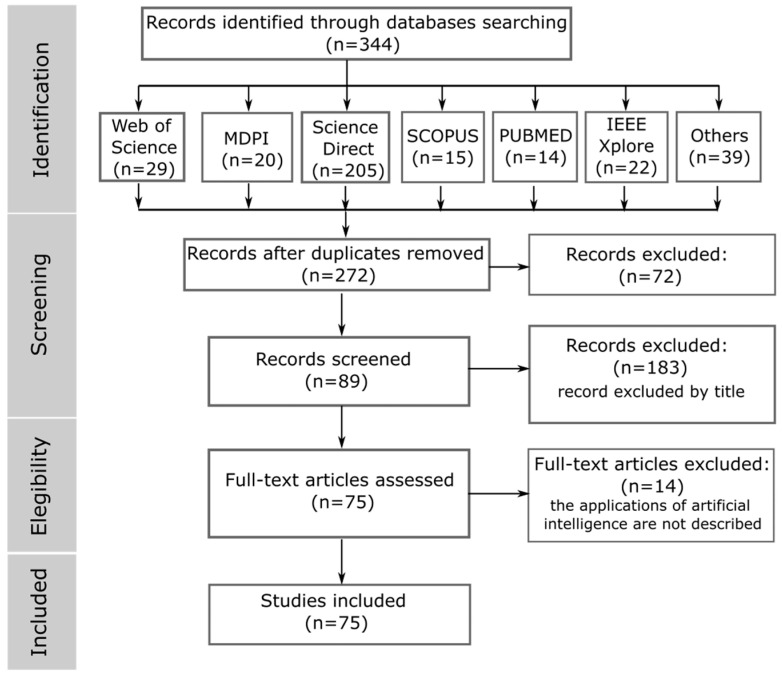
Workflow in the systematic review with the PRISMA^®^ methodology.

**Figure 9 bioengineering-10-01039-f009:**
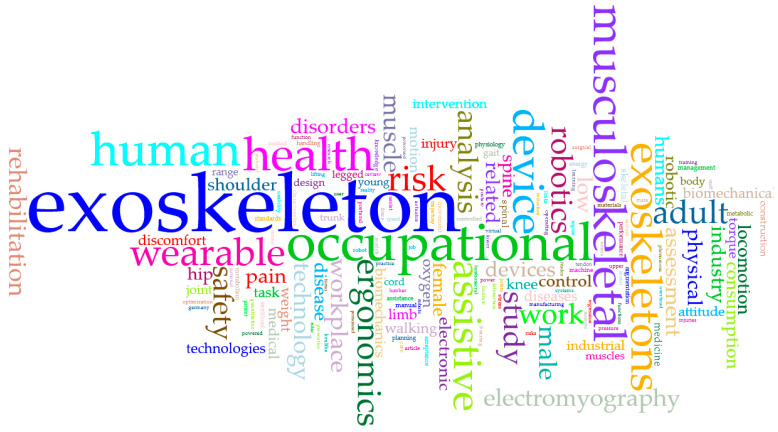
Word cloud from keywords with the use of the Voyant Tools^®^ tool.

**Table 1 bioengineering-10-01039-t001:** Examples of exoskeletons for assisting body parts during physically demanding tasks.

**Exoskeletons Shoulder-Arm Assistance**					
**Type**	**Exoskeleton**	**Muscle Activity** **and Movement**	**Joints’ Activity**	**Natural** **Movement**	**Lifting Speed** **on Arm**
Passive	EKsoVest	X			
Passive	H-VEX	X			
Passive	Fawcett Exsovestᵀᴹ	X			
Passive	Hombrox	X			
Passive	PAEXO		X	X	
Passive	EXHAUST	X			
Passive	SkelEx	X			
Active	EcoPick	X			
Active	Lucy	X			X
Active	MuscleSuite	X			
**Exoskeletons Back Assistance**					
**Type**	**Exoskeleton**	**Muscle Activity** **and Movement**	**Lower Energy** **Expenditure**	**Lower Thoracic Strength**	**Less Strength** **in the Lower Back**
Passive	BackX	X			
Passive	BNDR	X			
Passive	Laevo	X	X		
Passive	PLAD	X		X	X
Active	SPEDOR	X	X		X
Active	HAL	X	X		
Active	MeBot-EXO		X	X	X
Active	MK2b				X
**Exoskeletons Full-Body Assistance**					
**Type**	**Exoskeleton**	**Load Resistance**	**Arm** **Support**	**Physical** **Demands**	**Task** **Difficulty**
Active	Fortisᵀᴹ	X	X	X	X
**Exoskeleton Leg Assistance**					
**Type**	**Exoskeleton**	**Muscle Activity** **and Movement**	**Comfort Foot Position**	**Maintain** **Balance**	**Torsion** **Reduction**
Passive	LegX	X			
Passive	Silla sin silla	X	X		
Active	Traje de ANGEL	X		X	
Active	UMEx-oLEA	X			X

**Table 2 bioengineering-10-01039-t002:** Strings used in the search for scientific literature.

Database	String Search	Study Number
IEEE	ALL (exoskeletons AND occupational AND safety AND health)	22
Others	TITLE (exoskeletons AND occupational AND safety AND health)	39
Proquest	(exoskeleton) AND (occupational) AND (safety) AND (health)	20
PubMed	TITLE-ABS-KEY (exoskeletons AND occupational AND safety AND health)	14
Science Direct	All (exoskeletons AND occupational AND safety AND health)	205
Scopus	TITLE-ABS-KEY (exoskeletons AND occupational AND safety AND health)	15
Web of Science	All Fields (exoskeletons AND occupational AND safety AND health)	29
	Total number of studies	344

**Table 3 bioengineering-10-01039-t003:** Quality assessment questions.

N°	Quality Assessment Questions	Answer
QA1	Does the paper describe devices or technologies for safety and occupational health?	(+1) Yes/(+0) No
QA2	Does the document specify how technology improves working conditions?	(+1) Yes/(+0) No
QA3	Does the paper describe the principles and technical characteristics of the operation of these technologies?	(+1) Yes/(+0) No
QA4	Are the limitations of using these technologies described in the paper?	(+1) Yes/(+0) No
QA5	Is the journal or conference in which the paper was published indexed in SJR?	(+1) if it is ranked Q1, (+0.75) if it is ranked Q2, (+0.50) if it is ranked Q3, (+0.25) if it is ranked Q4, (+0.0) if it is not ranked.

## Data Availability

Not applicable.

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
