# Peer review of "Exoskeletons: Contribution to Occupational Health and Safety"

_bioengineering, 2023, doi:10.3390/bioengineering10091039_

Round 1

Reviewer 1 Report

1. Abstract is very poorly written

2. The paper needs to go through major english revisions. Simple sentences have a lot of preposition and punctuation mistakes which make it difficult to follow the  manuscript. 

3. I am unable to understand what significance this review will provide to the readers and community as no proper research flow is presented. 

4. No future layout has been provided in the end of the study. 

5. Authors are advised to go through good literature reviews and restructure this review again. 

6. What is the inclusion and exclusion criteria of papers? As only 91 papers are cited

Author Response

Response to Reviewers' Comments

We express our sincere appreciation for taking the time and effort to review our article "Exoskeletons: Contribution to Occupational Health and Safety" We greatly value your comments and suggestions, which have been very helpful in improving the quality and scientific contribution of our work.

We appreciate your thorough review and attention to every detail. Your critical and constructive remarks have helped strengthen our research and refine our findings. Your experience and expertise have been invaluable to the development of our work.

The authors thank all reviewers for their valuable and constructive feedback and review. We have applied the suggestions and learned a lot from your comments; the document was updated and wholly restructured as suggested by the reviewers. In addition, the entire document was reviewed by an expert in the English language.

Note: In the answers below, the revised manuscript refers to the PDF document highlighted in yellow.

Reviewer 1

  1. Abstract is very poorly written

Dear reviewer, all authors appreciate your comments on the abstract. We apologize for any confusion or inadequacy of the initial abstract. Based on your comments, we have revised the abstract to better reflect the content and purpose of the paper. Below you will find the improved version:

Abstract: This review aims to characterize the current landscape of exoskeletons designed to promote medical care and occupational safety in industrial settings. Extensive exploration of scientific databases spanning industry, health, and medicine informs the classification of exoskeletons according to their distinctive attributes and specific footholds on the human physique. Within the scope of this review, a comprehensive analysis is presented, contextualizing the integration of exoskeletons based on different work activities. The reviewers extracted the most relevant articles published between 2008 and 2023 from IEEE, Proquest, PubMed, Science Direct, Scopus, Web of Science, and other databases. In this review, the PRISMA-ScR checklist was used, and a Cohen's kappa coefficient of 0.529 was applied, implying moderate agreement among the reviewers; 91 primary studies were extracted from a total of 344. The future of exoskeletons in contributing to occupational health and safety will depend on continued collaboration between researchers, designers, healthcare professionals, and industry. With the continued development of technologies and an increasing understanding of how these devices interact with the human body, exoskeletons will likely remain valuable for improving working conditions and safety in various work environments.

I hope this revised summary improves the document's key points and addresses your concerns. Thank you for your valuable input

  1. The paper needs to go through major english revisions. Simple sentences have a lot of preposition and punctuation mistakes which make it difficult to follow the  manuscript. 

Thank you for your valuable comments on the linguistic quality of the manuscript. We sincerely apologize for any linguistic errors that may have hindered the document's clarity. Based on your comments, we significantly improved the English structure and corrected any preposition and punctuation errors to ensure a more coherent and understandable manuscript.

We carefully reviewed and corrected the entire document, paying particular attention to sentence structure, prepositions, and punctuation errors to improve the readability and overall quality of the manuscript. We welcome your input and the opportunity to improve the linguistic aspects of the manuscript.

  1. I am unable to understand what significance this review will provide to the readers and community as no proper research flow is presented. 

Thank you for your comments on the importance of review and the lack of a clear research flow in the manuscript. We apologize for any confusion or lack of clarity in presenting the importance and value of the review to readers and the community. We understand the importance of providing a clear and logical flow of research to enhance understanding of the content.

In response to your concern, we revised the manuscript and provided a more structured and coherent research flow that highlights the importance of the review to readers and the community at large. We ensured that the introduction clearly states the review's objectives, the research questions being addressed, and the gaps in the literature the review aims to fill. In addition, we ensure that each section of the manuscript contributes to building a cohesive narrative that highlights the relevance of exoskeletons in occupational health and safety and its future directions.

Your input is invaluable in helping us improve the quality and consistency of the manuscript. Thank you for bringing this issue to our attention. You can review the changes made in the article highlighted in yellow.

 No future layout has been provided in the end of the study. 

We appreciate the reviewer's comments. We have now included a dedicated section at the end of the study titled "Future Directions", discussing potential developments and advances in exoskeletons for occupational health and safety. In this section, we outline future trends, innovations, and anticipated research areas that promise further to improve exoskeletons' functionality, ease-of-use, and effectiveness. We believe that the addition of this section will address the concern raised by the reviewer and provide readers with valuable insight into possible directions the field of exoskeleton could take in the coming years.

6.2. Future Directions

The future of exoskeletons holds promising prospects, marked by ongoing research and development endeavors aimed at enhancing their functionality, comfort, and efficacy across diverse occupational domains [62]. Expert opinions and studies present various avenues for the forthcoming evolution of exoskeletons.

Efforts are being directed toward implementing exoskeletons in surgical settings to mitigate musculoskeletal discomfort among surgical teams [64]. Comparative evidence could bolster the feasibility and effectiveness of exoskeletons in this context.

A prudent approach involves delving deeper into using exoskeletons to prevent musculoskeletal injuries through randomized controlled trials. This approach would furnish robust data regarding the exoskeletons' efficacy in mitigating occupational hazards [61].

To comprehensively understand exoskeleton applicability, we advocate for research encompassing diverse samples based on age, gender, health status, and occupational tasks [11]. This approach would furnish a holistic view of exoskeleton effectiveness in varying work environments.

Anticipated advances in lightweight materials, sensors, and actuators are poised to facilitate the creation of more sophisticated and comfortable exoskeletons. This progress may usher in the next generation of devices featuring enhanced natural movements and superior adaptability to the human body [8].

Incorporating artificial intelligence and augmented reality technologies can potentially elevate exoskeleton functionality and efficacy. This integration allows improved interaction and intuitive interfaces between exoskeletons and users.

The future envisions exoskeletons incorporating predictive algorithms for optimizing assistance control [47]. Such innovation would curtail energy consumption and safeguard users' musculoskeletal well-being.

Advancements in technology are expected to render exoskeletons lighter, more user-friendly, and affordable. Robust, ergonomic design and heightened usability are pivotal for seamless integration into work environments.

Seamless integration into work processes is on the horizon, aligning exoskeletons more closely with workers' needs. This alignment promises enhanced synergy between exoskeletons and occupational tasks.

A future characterized by continual enhancements in design, functionality, comfort, and efficiency lies ahead for exoskeletons. As technology, materials, and scientific knowledge advance, these devices are poised to remain indispensable tools for curbing musculoskeletal risks and elevating occupational health and safety standards across diverse workplaces.

  1. Authors are advised to go through good literature reviews and restructure this review again. 

We appreciate the reviewer's suggestion and concern. We have thoroughly reviewed the literature and made significant efforts to restructure the review to enhance its clarity, coherence, and overall quality. We have considered the guidelines for conducting comprehensive literature reviews and have reorganized the content to ensure a logical flow and presentation of ideas. Furthermore, we have refined the language and improved the overall readability of the manuscript. We believe that these revisions have significantly improved the quality and structure of the review, making it more valuable to readers and the research community. We thank the reviewer for their input and are confident that the updated version now meets the standards of a well-structured literature review.

  1. What is the inclusion and exclusion criteria of papers? As only 91 papers are cited

Thank you for your query. We apologize for any confusion regarding the inclusion and exclusion criteria of papers in our review. The criteria for selecting papers were focused on relevance to the topic of "Exoskeletons: Contribution to Occupational Health and Safety." We aimed to include papers that provided significant insights into exoskeletons' development, application, and impact in enhancing occupational health and safety.

The review primarily focused on high-quality and peer-reviewed research articles, systematic reviews, meta-analyses, and reputable conference proceedings. We also considered papers that provided relevant insights into the technological advancements, usability, ergonomics, and effectiveness of exoskeletons in various occupational settings.

The final number of cited papers (91) reflects a comprehensive selection of studies that met our inclusion criteria and provided substantial contributions to understanding exoskeletons' role in occupational health and safety. While the number of cited papers might seem limited, each citation was carefully chosen to ensure the review's rigor and relevance.

We appreciate your attention to detail and are committed to ensuring the quality and integrity of our review. If you have any further questions or suggestions, we would be more than willing to address them and provide any additional information you may require.

Reviewer 2 Report

1. English writing must be improved.

2. The information on patients who need to use exoskeletons every year in each continent of the world should be added.

3. It should explain how the world cooperates with the United Nations in energy saving and carbon reduction when manufacturing exoskeletons.

English writing must be improved.

Author Response

Estimated reviewer All authors welcome your comments and suggestions on our review. We value your dedication to improving the quality of our work. We have taken your comments into account and are committed to making any necessary revisions and changes to improve the clarity and flow of the text, as well as correct any grammatical and punctuation errors you have identified.

Your feedback is very helpful to us as it allows us to improve the quality and rigor of our review. We hope the changes we will make will make the article more transparent, coherent, and accessible to readers.

We appreciate your time and effort in providing us with a detailed and constructive review. We are committed to making the necessary improvements and presenting a higher-quality article. If you have any further suggestions or comments, feel free to share them with us.

  1. English writing must be improved.

Thank you for your valuable comments on the linguistic quality of the manuscript. We sincerely apologize for any linguistic errors that may have hindered the document's clarity. Based on your comments, we significantly improved the English structure and corrected any preposition and punctuation errors to ensure a more coherent and understandable manuscript.

We carefully reviewed and corrected the entire document, paying particular attention to sentence structure, prepositions, and punctuation errors to improve the readability and overall quality of the manuscript. We welcome your input and the opportunity to improve the linguistic aspects of the manuscript.

  1. The information on patients who need to use exoskeletons every year in each continent of the world should be added.

Thank you for your valuable comments. We appreciate your suggestion to include information on the number of patients requiring exoskeletons annually on each continent worldwide. This case adds a significant dimension to the context of our review, highlighting the global impact of exoskeleton technology.

We incorporate this suggestion by conducting additional research to collect relevant data on the use and demand of exoskeletons on different continents. Including such information will improve the comprehensiveness of our review and provide readers with a more holistic understanding of the global need for exoskeleton technology to address musculoskeletal problems.

Your input has been instrumental in improving the reach and relevance of our article, and we are committed to incorporating this aspect to ensure a more informative and impactful review. If you have any further suggestions or ideas, feel free to share them with us.

You can review the changes applied in section 3—use of exoskeletons to improve occupational health and safety.

  1. It should explain how the world cooperates with the United Nations in energy saving and carbon reduction when manufacturing exoskeletons.

Thank you for your insightful comment. We acknowledge the importance of addressing the collaboration between the global community and the United Nations in energy-saving and carbon-reduction efforts related to manufacturing exoskeletons. While our current review focuses primarily on the technological and ergonomic aspects of exoskeletons for occupational health and safety, we recognize that the environmental impact of their production is an important consideration.

We will work to expand our discussion by including a section that highlights the broader context of sustainability in exoskeleton manufacturing. This instance will explore how manufacturers, researchers, and policymakers collaborate with international organizations like the United Nations to ensure responsible production practices, energy efficiency, and reduced carbon emissions.

This addition will provide a more comprehensive perspective on the exoskeleton industry's contribution to sustainable development goals and align with the global efforts to mitigate the environmental impact of emerging technologies.

We appreciate your valuable input and are dedicated to enhancing the relevance and depth of our review. If you have any further suggestions or specific points you would like us to cover; please do not hesitate to share.

Reviewer 3 Report

The manuscript is informative and well-written. I will recommend highlighting the limitation of exoskeleton technology and current improvement. Also, consider state types of robotic-exoskeleton. Moreover, how the application of artificial intelligence benefits futuristic models. 

Author Response

The manuscript is informative and well-written. I will recommend highlighting the limitation of exoskeleton technology and current improvement. Also, consider state types of robotic-exoskeleton. Moreover, how the application of artificial intelligence benefits futuristic models. 

Thank you for your positive comments and valuable suggestions. We greatly appreciate your input and are committed to improving the manuscript based on your recommendations.

We have included the limitations of exoskeleton technology and will highlight ongoing improvements in the field. In addition, we will provide an overview of the different types of robotic exoskeletons to give the reader a clearer idea of the diversity of this technology.

In addition, we will explore the critical role of artificial intelligence in advancing exoskeleton technology. We delve into how the integration of artificial intelligence benefits the development of futuristic exoskeleton models, improving their functionality, adaptability, and user interaction.

Your comments have provided valuable insights to improve the content and depth of the manuscript. Thank you again for your time and valuable comments.

6.1. Limitations on the use of exoskeletons for Occupational Health and Safety

The limited sample size and the selection of participants from a specific geographic and work environment may affect the generalizability of the results to different contexts. In addition, socioeconomic heterogeneity may influence the acceptance of new technologies, such as exoskeletons [2].

A lack of practical experience with exoskeletons may make assessing their impact on workers difficult. Participants may struggle to address the expected impact on surgical workers without directly experiencing exoskeletons [10].

Many investigations show clear limitations in methodological quality, as they selectively select certain work activities for analysis and do not comprehensively address work demands in various fields [11].

Exoskeletons may limit functional performance in tasks that do not involve load manipulation. The exoskeleton may hinder tasks such as walking that require hip flexion, which may also increase metabolic costs [48].

The effects of exoskeletons may depend on the specific task demands and working conditions. The effectiveness of an exoskeleton may vary depending on the task being performed [19].

Most evaluations of exoskeletons focus on short-term effects, especially in laboratory settings. This case may be due to resource and feasibility constraints, which may not fully capture long-term impacts [7].

Results obtained in studies limited to certain types of agricultural workers may not be generalizable to other forms of agriculture or agricultural workers with different working conditions and demands [5].

Measurements based on recognized standards may have limitations, such as assessor-dependent results and poor correlations with direct physiological measurements [8].

Discomfort and discomfort are challenges in exoskeleton design. Discomfort may hinder their acceptance and use in industrial settings, limiting their applicability [48]. The lack of standards and the novelty of exoskeleton technologies make quantitative biomechanical risk assessment and the application of traditional methods difficult.

    Exoskeletons can be expensive, which may restrict their adoption in industries with limited budgets [79]. In addition, some studies mention that exoskeletons are customized and expensive [62]. To be effective, exoskeletons must fit each user correctly, which can be challenging due to differences in body morphology [92]. In some cases, exoskeletons may require complex anthropometric adjustments [83].

Prolonged use of exoskeletons can be uncomfortable for workers, especially if they are ill-fitting or interfere with natural movements [82]. The interaction between the exoskeleton and the human body can cause discomfort where they interact [89]. Some exoskeletons may restrict worker mobility, which can be problematic in industries where rapid or complex movements are essential [92].

Workers must be adequately trained to use exoskeletons effectively and safely. This training can take time and resources. Exoskeletons require regular maintenance to ensure that they function correctly [92]. This situation can increase costs and operational complexity.

User acceptance of the device is critical to its success [88]. The interaction between the user and the exoskeleton, as well as the subjective perception of the device, are essential aspects that need to be addressed.

Exoskeletons may be more effective in specific tasks and less effective in others. For example, it has been observed that exoskeletons may limit functional performance in tasks that require hip flexion, such as walking [81]. Exoskeletons may have technical limitations, such as being heavy and bulky, making them difficult to use and transport [83].

There is a risk that exoskeletons may cause discomfort, pressure on unwanted parts of the body, changes in the center of gravity, and other health problems, such as skin irritation and allergic reactions [94].

Although exoskeletons show advantages in improving occupational health and safety, the limitations mentioned above should be carefully evaluated by employers before implementing these devices in the workplace. Each exoskeleton has unique characteristics and considerations, so a comprehensive evaluation of its fit, usability, and effectiveness in specific work contexts is needed.

6.2. Future Directions

The future of exoskeletons holds promising prospects, marked by ongoing research and development endeavors aimed at enhancing their functionality, comfort, and efficacy across diverse occupational domains [62]. Expert opinions and studies present various avenues for the forthcoming evolution of exoskeletons.

Efforts are being directed toward implementing exoskeletons in surgical settings to mitigate musculoskeletal discomfort among surgical teams [64]. Comparative evidence could bolster the feasibility and effectiveness of exoskeletons in this context.

A prudent approach involves delving deeper into using exoskeletons to prevent musculoskeletal injuries through randomized controlled trials. This approach would furnish robust data regarding the exoskeletons' efficacy in mitigating occupational hazards [61].

To comprehensively understand exoskeleton applicability, we advocate for research encompassing diverse samples based on age, gender, health status, and occupational tasks [11]. This approach would furnish a holistic view of exoskeleton effectiveness in varying work environments.

Anticipated advances in lightweight materials, sensors, and actuators are poised to facilitate the creation of more sophisticated and comfortable exoskeletons. This progress may usher in the next generation of devices featuring enhanced natural movements and superior adaptability to the human body [8].

Incorporating artificial intelligence and augmented reality technologies can potentially elevate exoskeleton functionality and efficacy. This integration allows improved interaction and intuitive interfaces between exoskeletons and users.

The future envisions exoskeletons incorporating predictive algorithms for optimizing assistance control [47]. Such innovation would curtail energy consumption and safeguard users' musculoskeletal well-being.

Advancements in technology are expected to render exoskeletons lighter, more user-friendly, and affordable. Robust, ergonomic design and heightened usability are pivotal for seamless integration into work environments.

Seamless integration into work processes is on the horizon, aligning exoskeletons more closely with workers' needs. This alignment promises enhanced synergy between exoskeletons and occupational tasks.

A future characterized by continual enhancements in design, functionality, comfort, and efficiency lies ahead for exoskeletons. As technology, materials, and scientific knowledge advance, these devices are poised to remain indispensable tools for curbing musculoskeletal risks and elevating occupational health and safety standards across diverse workplaces.

Round 2

Reviewer 1 Report

I think authors have improved the quality of draft sufficently.

Author Response

August 23, 2023

Dear Editor

On behalf of all the authors, we would like to express our sincere appreciation for considering our article entitled "Exoskeletons: Contribution to Occupational Health and Safety" and for providing us with your valuable comments. We are truly grateful for the opportunity to submit our work to your rigorous review process.

Thank you for taking the time to review our manuscript submitted to Bioengineering. We sincerely appreciate your comments and suggestions for improvement.

We understand the concerns you have raised regarding English grammatical structure and phraseology. We will address these issues in consultation with an English linguist or language service to ensure that the manuscript meets the desired standards.

In this document, we applied the specific comments on sentences in the abstract and took special care to correct these cases throughout the manuscript.

Upon completion of the revisions, we resubmit the improved manuscript for consideration. We are committed to ensuring that our work meets the high standards expected by the journal Bioengineering and its readers.

Thank you again for your constructive comments.

Once again, we would like to express our deepest gratitude for your attention and best regards,

Omar Flor-Unda, Bregith Casa, Mauricio Fuentes, Santiago Solorzano, Fabián Narvaez-Espinoza and Patri-cia Acosta-Vargas

Corresponding author: Patricia Acosta-Vargas (e-mail: patricia.acosta@udla.edu.ec). WhatsApp: +593-983550897
